# The Effect of Floor Height on Secondhand Smoke Transfer in Multiunit Housing

**DOI:** 10.3390/ijerph19073794

**Published:** 2022-03-23

**Authors:** Emily Gill, Elle Anastasiou, Albert Tovar, Donna Shelley, Ana Rule, Rui Chen, Lorna E. Thorpe, Terry Gordon

**Affiliations:** 1Department of Population Health, Grossman School of Medicine, New York University, New York, NY 10016, USA; elle.anastasiou@nyulangone.org (E.A.); albert.tovar@nyulangone.org (A.T.); lorna.thorpe@nyulangone.org (L.E.T.); terry.gordon@nyulangone.org (T.G.); 2Department of Public Health Policy and Management, School of Global Public Health, New York University, New York, NY 10003, USA; donna.shelley@nyu.edu; 3Department of Environmental Health and Engineering, Johns Hopkins Bloomberg School of Public Health, Baltimore, MD 21205, USA; arule1@jhu.edu (A.R.); rchen42@jhu.edu (R.C.)

**Keywords:** secondhand smoke, smoke transfer, airborne nicotine, multiunit housing, building height

## Abstract

Secondhand smoke (SHS) exposure remains a major public health concern in the United States. Homes have become the primary source of SHS exposure, with elevated risks for residents of multiunit housing. Though this differential risk is well-documented, little is known about whether SHS exposure varies by floor height. The aim of this study was to examine whether SHS accumulates in higher floors of multiunit housing. Using validated passive nicotine sampling monitors, we sampled air nicotine concentrations on multiple floors of 21 high-rise (>15 floors) buildings in New York City. Within the buildings, measurements were collected in three locations: non-smoking individual apartments, hallways and stairwells. Measurements were collected in two winter and two summer waves to account for potential seasonality effects. We analyzed the percent of filters with detectable nicotine and quantified nicotine concentration (µg/m^3^). Higher floor levels were positively associated with both airborne nicotine measures, with some variation by location and season observed. In winter, the trends were statistically significant in apartments (floors ≤7: 0.022 µg/m^3^; floors 8–14: 0.026 µg/m^3^; floors ≥15: 0.029 µg/m^3^; *p* = 0.011) and stairwells (floors ≤7: 0.18 µg/m^3^; floors 8–14: 0.19 µg/m^3^; floors ≥15: 0.59 µg/m^3^; *p* = 0.006). These findings can inform interventions to mitigate the SHS exposure of residents in multiunit housing.

## 1. Introduction

In the decades since the landmark 1964 Surgeon General report on smoking and health, smoking prevalence among adults in the United States (U.S.) has steadily declined, from more than 40% of U.S. adults in the 1960s to less than 20% in 2018 [1]. Despite this progress, tobacco smoke remains the leading cause of preventable death in the U.S., with secondhand smoke (SHS) exposure accounting for up to 41,000 deaths annually [2,3].

Smoke-free policies have been highly effective in reducing SHS exposure in public settings, such as bars and restaurants [3]. As a result, the home environment is now the major source of SHS exposure. Residents of multiunit housing (MUH) are especially at risk, as smoke can travel between individual residences and building common spaces [4]. A nationwide survey of U.S. adults revealed that 34.4% of MUH residents living in smoke-free homes still experience SHS incursions from sources outside of their own units [5].

Ventilation is a major factor in how SHS moves throughout MUH. Buildings are ventilated by natural air exchange (which results from pressure differences and air buoyancy), infiltration (uncontrolled airflow through cracks and leaks of the building) and mechanical ventilation (i.e., fans) [6]. These modes of air distribution create pockets of varying pressure and temperature, which in turn cause air to flow between and within the spaces of the building. Included in this is a phenomenon called the stack effect, wherein temperature differentials move air vertically inside the building: warm air rises, creating a powerful upward-flowing current [7,8,9]. 

Although there is an abundance of research on the stack effect and other vertical airflow phenomena in MUH, as well as an abundance of research on SHS exposure in MUH, studies examining whether SHS exposure risk increases at higher floor levels in MUH are lacking [4,5,6,7,8,9,10,11,12,13,14,15,16,17]. Tobacco smoke (a mixture of gases and particulate matter) is able to travel along surrounding airflows and generally rises, being warmer than the surrounding air [7,8,9]. In this brief analysis, we leveraged a large, longitudinal dataset of indoor air nicotine concentration data collected as part of a quasi-experimental study in 21 high-rise buildings (>15 floors) in New York City, to investigate whether SHS accumulates by floor. 

## 2. Materials and Methods

### 2.1. Study Population 

We selected a sample of 21 high-rise (>15 floors) buildings in New York City (NYC), comprised of 10 NYC Housing Authority (NYCHA) buildings and 11 private sector buildings which consisted of affordable or rent-assisted housing matched on population demographics. The buildings ranged from 21 to 44 stories high. The study protocol and procedures have been described elsewhere [18].

### 2.2. Air Nicotine Concentration

To measure airborne nicotine, we used passive bisulfate-coated filters which have been validated by previous studies as a low-cost, accurate method for quantifying environmental tobacco smoke [18,19]. Research staff placed filters in three different settings throughout the buildings: the individual apartments of non-smoking households, as well as building hallways and stairwells. The placement guidelines were uniform across the three settings and dictated that the filters be placed 1–2 m from the floor and at least 1 m away from a window or ventilation system, while also avoiding ‘dead zones’ such as corners [18]. As a quality control measure, we deployed duplicate filters for 2% of our samples. Duplicates were placed next to sample filters for the duration of the collection period, and their results were then compared against those of the sample filters to measure the uncertainty of our method. Four waves of data were collected in total, with two summer (April–September, 2018–2019) and two winter (December–March, 2018–2020) waves to account for potential seasonal influence. The filters were prepared and analyzed at Johns Hopkins University Bloomberg School of Public Health using the school’s Secondhand Smoke Exposure Assessment Laboratory standard operating procedures. The nicotine limit of detection (LOD) for the filters was 0.017 µg/m^3^ [20]. 

### 2.3. Data Analysis

Air nicotine concentrations were calculated using geometric means, and the percentage of filters with detectable nicotine were estimated using the LOD. We used descriptive statistics to characterize nicotine exposure stratified by floor groupings, which consisted of floors 7 and lower, floors 8 through 14, and floors 15 and above. Floor groupings were chosen using NYC building code, which defines a high-rise as having occupied floors more than 75 feet above street level, or roughly 7 floors. Differences between floor groupings were compared using a chi-square test of independence or a one-way analysis of variance. Last, we performed a sensitivity analysis using simple linear regression to determine if floor height was a significant predictor of nicotine concentration. A regression coefficient and coefficient of determination, or R^2^, were calculated for each unit type and season. All estimates were configured to be statistically significant (*p* < 0.05). Analyses were conducted on Stata 16.1. (StataCorp LLC, College Station, TX, USA).

## 3. Results

A total of 1198 measurements were collected over the study period. Table 1 presents the resultant nicotine concentrations across different floor groups for each unit setting (non-smoking apartments, stairwells and hallways) and season. A comparison of nicotine concentration across the floor groups showed some significant differences. As floor height increased, apartments in winter showed an increase in both nicotine concentration (floors ≤7: 0.022 µg/m^3^ (95% CI, 0.019–0.025); floors 8–14: 0.026 µg/m^3^ (95% CI, 0.022–0.031); floors ≥15: 0.029 µg/m^3^ [95% CI, 0.025–0.033] *p* = 0.011) and the proportion of filters with detectable nicotine (floors ≤7: 14.6%; floors 8–14: 21.9%; floors ≥15: 28.3%, *p* = 0.041) (Table 1). Stairwells followed the same general trend for both summer and winter. Stairwells in summer showed an increase in nicotine concentration across the three floor groups (0.13 µg/m^3^ (95% CI 0.05–0.35), 0.28 µg/m^3^ (95% CI 0.17–0.45), and 0.35 µg/m^3^ (95% CI 0.23–0.512), *p* = 0.203 from lowest to highest, respectively). The proportion of filters with detectable nicotine also increased in tandem with floor height (72.7%, 89.5%, 96.9%, *p* = 0.067). Stairwells in winter followed the same trend for nicotine concentration (0.18 (95% CI 0.06–0.53), 0.19 (95% CI 0.11–0.33), 0.59 (95% CI 0.37–0.93), *p* = 0.006), though filters above LOD did show a slight dip in the second highest floor group (80%, 75.7%, 93.3%, *p* = 0.153). For hallways, there was no discernible pattern in geometric means, and similarly the proportion of filters with detectable nicotine did not follow any particular trend, irrespective of season. The trends remained similar when aggregated by season (Appendix A). 

Despite these trends, the R^2^ was quite low for all locations and seasons (Table 2). In addition, the measurement of geometric means did yield high variability, in some cases having ranges from the LOD (0.017 µg/m^3^) to 5.903 µg/m^3^, as with the winter stairwell results (Figure 1). Last, the coefficient of variation generated using our duplicate sample results indicated only 8% variability, suggesting that the results from filters were reliable. 

## 4. Discussion

To the best of our knowledge, this is the first study examining the effect of floor height on airborne nicotine concentration in high-rise MUH. As per the literature, we expected to see an accumulation of nicotine in higher levels of the buildings, given that nicotine is expelled in both the gaseous and particle phases of secondhand cigarette smoke and has an ability to ride prevailing airflow, which moves vertically throughout high-rise buildings [7,9,10].

The fact that we saw some evidence of this effect, specifically in apartments and stairwells, could be a true indication of a rise in SHS with increasing floor level. That the trends were mostly consistent across two different measurements, both concentration and the presence of airborne nicotine, adds credibility. Where trends in SHS were seen, the results were often significant or nearly significant. In addition, based on our assessment of the variability of the findings compared to the duplicate filters, we have confidence that the results reflect true differences. Finally, our nicotine concentrations were in line with those seen in similar settings using comparable methods of measurement, as was our more general finding that SHS exposure is building-wide, including non-smoking homes and especially in building common areas [11,12].

Other studies examining air flow in MUH have implicated doors as the primary source of air transfer within buildings [4,10]. McKeen and Liao demonstrated that hallways in particular contain myriad sources of leakage as a result of their proximity to both individual units and building shafts, and that corridor air flow is highly variable depending on ever-shifting pressure differentials within the building, airtightness, and outside temperature [10]. These factors may explain why we did not see a discernible pattern among our sample hallways, which had exposure to at least twelve apartment unit doors, two stairwells, and two elevators, and occasionally direct exposure to the outdoors in the form of windows or terraces. 

Regarding seasonality and airborne nicotine, our findings were mostly consistent with those of previous literature. In the winter months, the stack effect within buildings becomes more pronounced as the temperature divergence between the inside and outside of the building is increased [9,10]. To that end, we expected to see more pronounced trends in the winter wave of our data collection, as was the case in our apartment results, which showed a pattern of increasing concentration with floor height in winter, as well as stairwells, where the increase in airborne nicotine concentration was statistically significant. However, due to the increased likelihood that smokers choose to stay indoors to smoke, as well as decreased ventilation with the outdoors, we expected to see increased concentrations in the winter months, which, apart from a modest increase seen in apartments, did not clearly appear in our data [10,11,13,14]. In the summer months, the use of air conditioners can create a reverse stack effect wherein air flows downward through the building. Though less powerful than the normal stack effect, the reverse stack is also a result of temperature differences between the inside and outside air [9,15]. As none of our study sites had building-wide cooling systems, it is possible that the reverse stack would primarily affect apartments where air conditioners were located. This may explain why a clear trend existed in our winter apartment results, but not in summer.

Despite our results, we cannot completely disregard the possibility of chance in the effect of floor height on airborne nicotine concentration, especially when considering the R^2^ seen in the sensitivity analysis. Possible sources of obfuscation include the sample sizes and the variation in measurements. Though the overall study had a large sample size, the number decreased when divided into categories, as can be seen for stairwells and hallways. Moreover, the within-category variation was quite large in some cases. 

SHS transfer is dependent on many variables, including building ventilation, the built environment, the outside environment, and human behavior. One limitation of this study is that we did not have background information or data for these extraneous variables that may confound or otherwise affect measurements. An additional limitation of the study could be that the specific selection of buildings and settings (e.g., urban and restricted to U.S.) may mean that the results are not generalizable. The strengths of our study include the use of validated passive nicotine filters to measure airborne nicotine, allowing for a reliable, objective measure of SHS [21,22]. In addition, we examined exposure in multiple buildings, as well as multiple settings within those buildings, allowing for a large and varied sample. 

## 5. Conclusions

Our investigation into the relationship between floor height and SHS exposure in high-rise MUH found evidence of the accumulation of SHS at higher floors. We also saw high levels of SHS exposure in common areas overall. Further studies are needed to confirm the varying risk of SHS exposure in high-rise settings and to evaluate whether this exposure can be mitigated. While strategies to reduce SHS incursions in occupied buildings exist, these treatments are imperfect, logistically difficult and costly, thus severely limiting their feasibility in low-funding contexts such as public housing [16,17]. Given the demonstrable risk of SHS exposure in MUH and in high-rises in particular, the importance of smoke-free housing (SFH) policies is quite clear. Until recently, SFH policies remained largely a voluntary option for building managers, with few examples of enacted policies at the state, local or federal level [23]. In 2018, the U.S. Department of Housing and Urban Development implemented a rule requiring all public housing authorities (PHAs) to implement smoke-free housing policies. Studies are ongoing to document the protective effect these rules have on PHA residents, a population that is at disproportionate risk of SHS exposure due to socioeconomic factors and their residence within MUH. 

## Figures and Tables

**Figure 1 ijerph-19-03794-f001:**
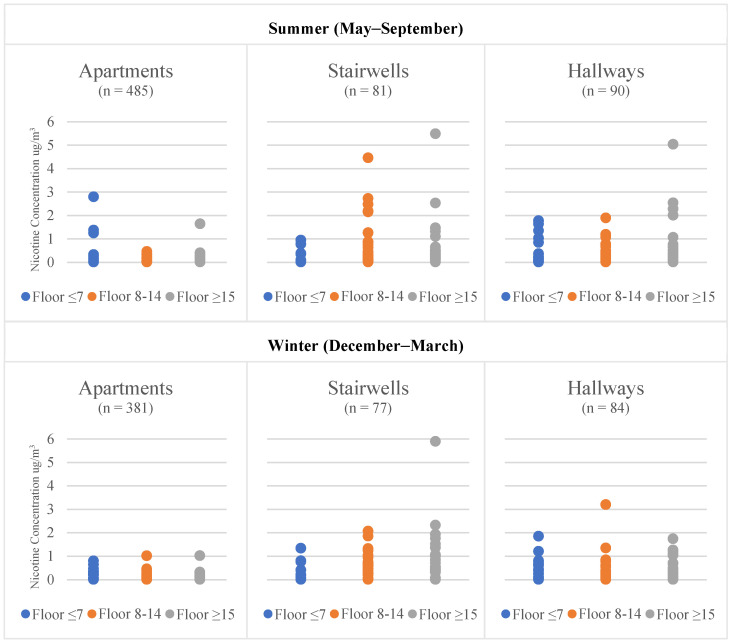
Variance in Air Nicotine Concentration by Location and Season.

**Table 1 ijerph-19-03794-t001:** Descriptive statistics.

Air Nicotine Concentration, 21 Low-Income High-Rise Buildings in NYC—By Location and Season								
	Summer (May–September)				Winter (December–March)			
	N	M (SD) or %	95% CI	*p*-Value	N	M (SD) or %	95% CI	*p*-Value
Apartments								
Floors ≤ 7	110				89			
Geometric Mean *		0.023 (2.524)	[0.020–0.028]	0.653		0.022 (1.993)	[0.019–0.025]	0.011
% above LOD **		15.5		0.857		14.6		0.041
Floors 8–14	162				119			
Geometric Mean		0.022 (1.929)	[0.019–0.024]			0.026 (2.470)	[0.022–0.031]	
% above LOD		13.6				21.9		
Floors ≥ 15	213				173			
Geometric Mean		0.022 (2.062)	[0.020–0.025]			0.029 (2.478)	[0.025–0.033]	
% above LOD		15.5				28.3		
Stairwells								
Floors ≤ 7	11				10			
Geometric Mean		0.13 (4.62)	[0.06–0.35]	0.203		0.18 (4.73)	[0.06–0.53]	0.006
% above LOD		72.7		0.067		80.0		0.153
Floors 8–14	38				37			
Geometric Mean		0.28 (4.51)	[0.17–0.45]			0.19 (5.20)	[0.11–0.33]	
% above LOD		89.5				75.7		
Floors ≥ 15	32				30			
Geometric Mean		0.35 (3.05)	[0.23–0.52]			0.59 (3.42)	[0.37–0.93]	
% above LOD		96.9				93.3		
Hallways								
Floors ≤ 7	23				23			
Geometric Mean		0.15 (5.37)	[0.08–0.32]	0.677		0.20 (4.73)	[0.10–0.39]	0.597
% above LOD		73.9		0.859		82.6		0.181
Floors 8–14	37				31			
Geometric Mean		0.14 (4.17)	[0.09–0.22]			0.20 (4.16)	[0.12–0.34]	
% above LOD		75.7				80.7		
Floors ≥ 15	30				30			
Geometric Mean		0.19 (5.15)	[0.10–0.35]			0.14 (6.12)	[0.07–0.27]	
% above LOD		80.0				63.3		

Abbreviations: LOD, limit of detection. * Geometric mean refers to airborne nicotine concentration measured in µg/m^3^. ** % above LOD refers to the proportion of filters with detectable nicotine.

**Table 2 ijerph-19-03794-t002:** Sensitivity Analysis Based on Coefficient of Determination.

	Summer	Winter
	Apartments	
Regression Coefficient	0.994	1.014
R^2^	0.004	0.015
	Stairwells	
Regression Coefficient	1.011	1.043
R^2^	0.007	0.070
	Hallways	
Regression Coefficient	1.023	1.002
R^2^	0.015	0.000

## Data Availability

The data that support the findings of this study are available from the corresponding author upon reasonable request.

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
