# Peer review of "The Effect of Floor Height on Secondhand Smoke Transfer in Multiunit Housing"

_ijerph, 2022, doi:10.3390/ijerph19073794_

Round 1

Reviewer 1 Report

“there is an abundance of literature on the stack effect and other vertical 51 airflow phenomena in MUH, as well as an abundance of literature on SHS exposure in 52 MUH”. A short list of literature should be added here.

Line 124: Can be specific about what you are concluding here. This might be the case in apartments during the winder, but not across all three types of locations and in both seasons. Actually the results indicate a counter conclusion for Hallways in the winter. The difference between hallways and apartments can be further discussed. 

Line 149: It seems high level (>7) apartments have higher nicotine concentrations during the winter. Could be worth checking here.

Even not significant, analysis should be made in aggregated levels (three location types aggregated or both season aggregated) and reported in the methods and results section. 

The study examined novel research hypotheses about the association between floor level and nicotine exposure. Although the study questions and findings are new, discussion about the importance of study findings is limited. 

Reviewer 2 Report

This manuscript describes a study on the effect of floor height on secondhand smoke transfer in multiunit housing. The study was done in New York in 21 high rise buildings y placing overall 1198 measurement filters in places determined by the study protocol. The topic of SHS in multiunit housing is of obvious public health importance.

The introduction mentions briefly the importance of SHS exposure and the problem of home environment and SHS. The methods part describes well the study design, the placement of the measurement filters and the data analyses. The results are presented in straight forward matter.

The text says that this study material was collected as part of a quasi-experimental study. It is nor quite clear whether this refers to this study (that to me is observatory) or another truely experimental study. Obviously a truely experimental study would be much more interesting and useful. This should be discussed  in the discussion part, because this observational study does show the existance of SHS in MUH as expected, but the effect of floor height, obviously existing, are not very substancial.

The importance of smokefree housing policies is clear and the policy possibilities for this are not topic of this article. Another issue is that this study is restricted to US situation. It is quite obvious that the problem of SHS is much greater the the tall multiunit housing buildings in the big cities of low and middle income countries.

Reviewer 3 Report

The aim of this study is to examine whether secondhad smoke accumulates in higher floors of multiunit housing. Passive nicotine sampling monitors were used in order to assess nicotine concentrations on multiple floors of New York City buildings. This paper address a relevant subject; however there are some aspects that need a deeper analysis. I consider that this paper require a major revision, according to the following remarks, before acceptance.

  • In the section 2.1 is made reference to “residents receive Section 8 subsidy vouchers”. This characterization is meaningless for people outside US. It is relevant to define better it’s meaning.
  • In the section “2.2. Air Nicotine Concentration”, the assessment of the uncertainty measurement of the air nicotine concentration shall be presented.
  • It is not clear that “% above LOD“ in table 1 means “proportion of filters with detectable nicotine”. The authors shall make this meaning clear.
  • The discussion shall consider the measurement uncertainty assessment.
  • The distribution of the smoke depends also on the location of the sources (active smokers) and on the external wind effect. Mention to how these two effects have been considered and may influence the measurement results shall be made.

Round 2

Reviewer 3 Report

No comments.